# Attraction of *Frankliniella occidentalis* Females towards the Aggregation Pheromone Neryl (*S*)-2-Methylbutanoate and Kairomones in a Y-Olfactometer

**DOI:** 10.3390/insects14060562

**Published:** 2023-06-16

**Authors:** Charles J. F. Chappuis, Marilyn Cléroux, Corentin Descombes, Yannick Barth, François Lefort

**Affiliations:** 1Analytical Chemistry Group, Haute École de Viticulture et Oenologie de Changins, HES-SO University of Applied Sciences and Arts Western Switzerland, 1260 Nyon, Switzerland; 2Plants and Pathogens Group, Research Institute Land Nature Environment, Geneva School of Engineering Architecture and Landscape, HES-SO University of Applied Sciences and Arts Western Switzerland, 1254 Jussy, Switzerland

**Keywords:** western flower thrips, pest management, olfaction, gas-phase concentration, olfactometer

## Abstract

**Simple Summary:**

Understanding insect olfactory perception can lead to more effective and specific ways to manage pests. In fact, odors have two components: (1) the quality, defined as its chemical composition, and (2) the quantity, defined by the concentration of the odor. The quantitative aspect is often missing and we aim to provide valuable information on the quantitative aspect of the odors that attract the western flower thrips, an insect that damages crops. In this study, we tested how the western flower thrips responded to different odors in an olfactometer. We measured the odor concentrations carried by the air and found that the thrips were most attracted to a pheromone at very low concentrations, much lower than the other odors we tested. Our results show that the attraction of western flower thrips to an odor depends on its quantitative aspect and could help to develop better ways of managing this pest.

**Abstract:**

An understanding of insect olfaction allows for more specific alternative methods of pest control. We evaluated the responses of the western flower thrips (WFT, *Frankliniella occidentalis*) in a Y-olfactometer to estimate gas-phase concentrations of the aggregation pheromone neryl (*S*)-2-methylbutanoate and known kairomones such as methyl isonicotinate, (*S*)-(-)-verbenone, and *p*-anisaldehyde. The gas-phase concentrations of these compounds were obtained from the release rates measured in dynamic headspace cells. The compounds were collected from the headspace using dried solid-phase extraction (SPE) cartridges and analyzed with a triple quadrupole GC-MS/MS. We observed that the aggregation pheromone significantly attracted WFT females at doses of 10 and 100 µg, whereas methyl isonicotinate and *p*-anisaldehyde significantly attracted WFT females at the highest dose. Verbenone did not produce any significant results. A completely different picture was obtained when the gas-phase concentrations were considered. The minimal gas-phase concentrations of the pheromone required to attract WFT females was 0.027 ng/mL, at least 100 times lower than that of the other two compounds. The relevance and implications of our results are discussed in light of the insect’s biology and pest management methods.

## 1. Introduction

Western flower thrips (WFT), *Frankliniella occidentalis* (Pergande) (Thysanoptera: Thripidae), are an economically important pest that attack a wide range of horticultural and agricultural crops. They cause damage to plants by feeding, ovipositing, and transmitting viruses, resulting in a significant decrease in crop yield and quality, leading to substantial economic losses [1]. Insecticides have been the most widely used method for controlling WFT populations. However, the development of insecticide resistance in WFT has raised concerns [2,3]. Consequently, scientists have been exploring alternative methods to control WFT populations, which are reviewed by Mouden et al. [4]. These methods include cultural control with trapping crops, activating the plants’ natural defense mechanisms, manipulating the plants’ genomes to increase their resistance to WFT, using natural enemies (such as entomopathogenic fungi, parasitoids, and predators), and behavioral control using visual and chemical signals.

Behavioral control has gained significant interest since the beginning of the century [5]. WFT can be manipulated using visual and/or chemical signals that they exploit to orientate, feed, reproduce, and communicate. For instance, yellow-colored traps act as a visual signal that attract WFT. When these traps are baited with an attractant as a chemical signal, more WFT are recruited from a distance, increasing the number of WFT caught by the yellow traps [6,7]. Chemical stimuli play an essential role in many strategies to monitor and control WFT populations. As attractants, they increase traps efficiency, allowing for “mass trapping” [8,9]. When combined with natural enemies, attractants not only recruit more WFT but also their predators [10]. Attractants are also used in the “lure and infect” strategy, where WFT are attracted and infected with entomopathogenic fungi [11]. Recently, a successful “push and pull” strategy used a combination of compounds mediating aggregation and alarm signals [12]. Therefore, researching the chemical communication of WFT is critical in developing efficient methods to control WFT populations.

Previous research has shed light on the chemical signals utilized by WFT to feed, find host plants, and to communicate with each other. The headspace volatiles of male WFT were analyzed, which led to the identification of two compounds, neryl (*S*)-2-methylbutanoate and (*R*)-lavandulyl acetate, that are involved in the aggregation behavior of WFT [13]. While the role of (*R*)-lavandulyl acetate is not completely understood and did not attract WFT in field trials, synthetic neryl (*S*)-2-methylbutanoate was effective in attracting both male and female WFT in the field [13]. This latter compound is considered a gold standard for controlling and monitoring WFT populations and is formulated in the commercial product ThripLine ams^®^ [9]. In addition, 7-methyltricosan, a hydrocarbon found in the cuticle of male WFT, was identified as a contact pheromone that causes females to remain close to the source and induces aggressive behavior in males upon contact [14]. The larvae of WFT also release an alarm pheromone composed of a mixture of decyl acetate and dodecyl acetate, which repels both adult and larval WFT and reduces oviposition [15]. This alarm pheromone also decreases landing rates and increases take-off rates [16].

WFT can use various chemical signals emitted by plants, likely due to the wide range of plants it feeds on and lays eggs in. Currently, 39 compounds have been tested as attractants [5]. Monoterpenoids such as geraniol, linalool, nerol, and β-citronellol, along with a sesquiterpenoid, β-farnesene, have been reported as attractants in a Y-olfactometer [17]. (*S*)-(-)-Verbenone, a monoterpenoid, was also found to be an attractant in a Y-olfactometer and elicited electroantennogram responses [18]. WFT are also attracted to aromatic compounds, such as benzaldehyde and *o*- and *p*-anisaldehyde, in a Y-olfactometer [17]. *p*-Anisaldehyde was also found to be attractive in the field [19]. Lactones, such as γ and δ-decalactone, attracted WFT in the field [20]. Pyridyl esters, a class of compounds that are seldom found in plants, including methyl isonicotinate, ethyl isonicotinate, and propyl isonicotinate, are highly attractive to WFT in Y-olfactometer studies [21]. Methyl isonicotinate is a gold standard for attracting WFT and other thrips species in the field, and it is included in the commercial product Lurem-TR^®^ [10,22].

In studies evaluating the response of WFT to volatile compounds, passive release systems were used to deliver them. Volatile compounds were loaded onto a matrix made of cotton, rubber, cellulose, or plastic, or were used neat or diluted in a sachet of various materials. These compounds evaporated from the matrix and were carried by air. The WFT were then exposed to the air carrying the olfactory stimuli. However, the gas-phase concentrations of the volatile compounds that stimulate the WFT were not known in most studies. Insects likely respond to a part of the amount of compounds that evaporate and are carried by the air, and not to the total amount of compounds that is loaded on a matrix such as a filter paper. This knowledge gap is significant when comparing the performance of different compounds in attracting insects and comparing the different responses of insects in various experimental setups. For example, in a recent study, WFT responses to methyl isonicotinate in a leaf disc experimental setup were different from those obtained in wind tunnels and field experiments [23]. The authors have highlighted the importance of determining the gas-phase concentrations in understanding WFT responses to volatile compounds. Moreover, exposing insects to known gas-phase concentrations is recognized as a major challenge in the development of efficient pest control methods, and the use volatile compounds and research on miniaturized sensors, as well as odor-plume modelling, is needed [24,25]. However, measuring the gas-phase concentrations of olfactory stimuli is challenging, especially in the field due to constantly changing environmental factors, turbulent airflows, and low concentrations of volatile compounds [26].

The resulting gas-phase concentration of a volatile compound evaporating from a passive release system is a combination of the air flow rate and the release rate. The release rate is influenced by the physical and chemical properties of both the volatile compound and the matrix, such as vapor pressure, stability, pH, surface, and sorption, as well as environmental factors such as temperature, air pressure, and flow. Nielsen et al. [27] investigated the factors that influence release rates and measured the release rates of methyl isonicotinate, ethyl isonicotinate, and propyl isonicotinate from different release systems at varying temperatures and air flows. Although this study provides valuable information for deploying such release systems efficiently in the field, it does not provide information on the gas-phase concentrations relevant to the WFT. 

In this study, we aimed to combine a behavioral assay with analytical chemistry in a laboratory setting that provided stable temperatures and controlled airflow. Our objective was to evaluate, using a Y-olfactometer, the responses of WFT to the aggregation pheromones, specifically (*S*)-2-methylbutanoate, and kairomones, including methyl isonicotinate *p*-anisaldehyde, and (*S*)-(-)-verbenone. We also measured the release rates of each compound using headspace cells and coupled gas chromatography–mass spectrometry (GC–MS) [28,29], under similar conditions, including temperature, airflow, and filter paper loading. The resulting gas-phase concentrations were calculated based on the measured release rates and the airflow in the olfactometer. We assessed the performance of each compound in attracting WFT based on their release rates and resulting gas-phase concentrations. As the commercially available (*S*)-2-methylbutanoate was not suitable for our study, we synthesized and characterized it using GC–MS and NMR analysis.

## 2. Materials and Methods

### 2.1. Chemicals

Chemicals, such as acetone (purity ≥ 99.8%), dichloromethane (99.9%), ethyl acetate (99.9%), and n-hexane (99%), were purchased from Carl Roth (Arlesheim, Basel-Landschaft, Switzerland). (*S*)-2-methylbutanoic acid (98%), *N*,*N*’-dicyclohexylcarbodiimide (DCC, 99%), diethyl ether (stabilized, 99.8%), 4-*N*,*N*’-diméthylaminopyridine (DMAP, 99%), HCl (Titrisol, Buenos Aires, Argentina), magnesium sulfate (99.5%), methyl isonicotinate (98%), neryl acetate (97%), *p*-anisaldehyde (98%), petroleum benzine (b.p. 40–60 °C), (*S*)-(-)-verbenone (94%), sodium hydrogenocarbonate (99.7%), SilicaGel (230–400 mesh), and activated charcoal (for GC) were purchased from Sigma–Aldrich (Buchs, Saint-Gallen, Switzerland). Solid-phase extraction (SPE) cartridges OASIS HLB 200 mg were purchased from Waters (Baden-Dättwil, Aargau, Switzerland).

### 2.2. Pheromone Synthesis

The aggregation pheromone of the WFT, neryl (*S*)-2-methylbutanoate, was synthetized according to the procedure described in Hamilton et al. [13]. Amounts of 0.061 g of 4-*N*,*N*’-dimethylaminopyridine (DMAP), 0.77 g of nerol, and 0.61 g of (*S*)-2-methylbutanoic acid were dissolved in 10 mL of dichloromethane and introduced in a 50 mL round-bottom flask. The flask was placed in an ice bath. The mixture was gently stirred and 1.24 g of *N*,*N*’-dicyclohexylcarbodiimide (DCC) were added step by step over 30 min. The round-bottom flask was removed from the ice bath and left at room temperature for 3 h to complete the reaction. The mixture was filtered on paper Whatman™ 589/2 (150 mm in diameter, Sigma, Buchs, Saint-Gallen, Switzerland) to remove the precipitate of *N*,*N’*-dicyclohexylurea. The precipitate was washed with petroleum spirit (b.p. 40–60 °C) and the filtrate was washed with a saturated solution of sodium hydrogenocarbonate followed by a HCl solution (0.1 M) and, finally, followed by deionized water. The organic phase was dried over magnesium sulfate and filtered. The pheromone was gently concentrated under nitrogen at 60 °C and purified on a column of 8 g of SilicaGel with a mixture of 15% of diethyl ether in petroleum spirit as eluent. The purified pheromone was then gently concentrated at 60 °C under nitrogen to remove the eluent. As the average yield was roughly 20%, we repeated the synthesis 4 times to obtain 1 g of pheromone. The structure of the pheromone synthetized was confirmed by NMR analysis (Appendix A). The purity of the pheromone synthetized was evaluated with the GC–MS described below; it was ≥96%. The linear retention indices (LRI) were measured on a polar column (DB-wax, 3 0 m × 0.25 mm, 0.25 µm, film thickness, Agilent Technologies, Santa Clara, CA, USA) and an apolar column (HP5, 30 m × 0.25 mm, 0.25 µm, film thickness, Agilent Technologies, USA) with the following method: helium flow 0.9 mL/min, split 10, 50 °C–5 min/10 °C/min–260 °C. The LRI were 1856 and 1576 on the polar and apolar columns, respectively.

### 2.3. Insects

The WFT colony was reared on *Phaseolus vulgaris* plants in Bugdorm 160 µm nylon mesh microcosms (60 cm × 60 cm × 60 cm) purchased from NHBS GmbH (Bonn, North Rhine-Westphalia, Germany) placed in a climate chamber (DR-36VL, Percival, Perry, IA, USA) with the following conditions: light cycle 16 h: 8 h, temperature of 25 °C, and 60–75% relative humidity. The WFT colony was established in 2021 with wild WFT collected in the Geneva area (Switzerland) on plants *Chrysenthemum* sp. and *Cannabis* sp. The thrips collected were morphologically and genetically identified as *Frankliniella occidentalis* by DNA sequencing of the amplified mitochondrial gene of the cytochrome oxidase unit 1 (CO1) gene (GenBank Accessions MZ519713 and MZ519714). The resulting sequences showed 100% identity over 637 bp and 635 bp with *F. occidentalis* COI sequences.

### 2.4. Behavioral Assay

The behavioral assay involved evaluating the response of adult female thrips to different olfactory stimuli using a Y-olfactometer following the procedure described in Koschier et al. [17]. The Y-olfactometer consisted of an 8 mm inner diameter glass Y-tube tilted at 25° and placed in a black box to prevent visual bias. A white LED lamp attached to the box illuminated the Y-tube with a 450 lux light intensity. The two arms of the Y-tube were connected via PTFE tubes to evaporation chambers that consisted of 0.1 L gas wash bottles, each containing a 1 cm^2^ filter paper loaded with 1 µL of the diluted compounds or the corresponding solvent. A clean air stream was generated by an air pump connected to a valve and an activated charcoal filter. The clean air stream was split and passed via PTFE tubes into the evaporation chambers. The wind speed was measured at the outlet of the main arm of the Y-tube with a hot wire anemometer (Sefram 9862, Saint-Etienne, France) and adjusted to 10 cm/s with the valve. The flow rate was calculated by multiplying the wind speed by the cross section of the Y-tube. The flow rate was roughly 300 mL/min in the main arm and 150 mL/min in each arm of the Y-tube. The temperature during the experiments was 25 °C. The olfactory stimuli were methyl isonicotinate, verbenone, and the pheromone diluted in hexane at 1 mg/mL, 10 mg/mL, and 100 mg/mL. With a 1 µL load, the corresponding masses were 1, 10, and 100 µg. *p*-Anisaldehyde was also used as an olfactory stimulus, but it was diluted in water at the same concentrations mentioned above, as it was not soluble in hexane. The aqueous solution of *p*-anisaldehyde was vigorously stirred to obtain an emulsion before pipetting. The filter papers were immediately transferred into the evaporation chambers when loaded and the olfactometer was flushed for 10 min to remove the solvent. The thrips were starved for 1 h in the dark at room temperature in 500 mL plastic boxes (Microbox^®^, Sac O2, Deinze, Belgium). Only one starved thrips was introduced at a time into the main arm of the Y-tube using a fine brush. The thrips chose between the odorant arm and the control arm at the junction of the Y-tube, and its choice was recorded when it reached the end of the odorant or control arm. If the thrips did not make a choice within 3 min, it was removed and recorded as undecided. Each experimental unit consisted of 25 thrips that chose one arm and this were repeated 3 times, resulting in 75 responding thrips per olfactory stimuli. The olfactometer was inverted every 5 thrips and the gas wash bottles were swapped between each repetition to avoid any positional bias. 

### 2.5. Headspace Cells

The method to collect volatile compounds from headspace cells was adapted from the one described in Herrmann et al. [28] and Starkenmann et al. [29]. Three headspace cells were built with 1 L gas wash bottles connected to air pumps (GilAir plus of Sensidyne, Lauper Instrument, Murten, Bern, Switzerland). The air was purified with activated charcoal and the flow rate in each cell was 200 mL/min (Figure 1). The volatile compounds were collected with SPE 200 mg Oasis HLB cartridges plugged at the outlet of the headspace cells and connected to the pumps (Figure 1). The cartridges were conditioned prior to the volatile collection with deionized water, acetone, and diethyl ether and dried at 50 °C for 1 h. The pheromone, methyl isonicotinate, *p*-anisaldehyde, and verbenone were prepared and loaded on filter papers as described earlier for the behavioral assay. The volatile compounds were collected for 50 min following a purge of 5 min. The temperature was 22 °C. The experiments were performed in triplicates and a blank control was carried out for each cell.

### 2.6. GC–MS/MS Analysis

The SPE cartridges were loaded with 100 µL of internal standard (0.1 mg/mL solution of neryl acetate in ethyl acetate) and desorbed with 1 mL of diethyl ether. The eluate was injected into a GC–MS for analysis. GC–MS analysis was performed on a 7890B gas chromatograph (Agilent Technologies, Santa Clara, CA, USA) coupled to a 7010 triple quadrupole mass spectrometer (Agilent Technologies, Santa Clara, CA, USA), equipped with a PAL autosampler MS-2000 (Bruker, Billerica, MA, USA). A split/splitless injector in splitless mode was used with the injector temperature at 250 °C. Separation was performed using a DB-wax capillary column (30 m × 0.25 mm, 0.25 µm, film thickness, Agilent Technologies, USA) and helium as carrier gas at a constant flow rate of 1.2 mL/min. GC oven was programmed as follows: 100 °C hold for 2 min followed by 3 °C/min increases up to 160 °C, then increased by 25 °C/min up to 230 °C. MS analysis was carried out with electron impact ionization operating at 70 eV and ion source was set at 230 °C. The acquisition was performed in MRM mode with the following transitions: 137→78 for methyl isonicotinate, 135→77 for *p*-anisaldehyde, 107→91 for verbenone, 93→51 for nerol, and 93→77 for the pheromone and neryl acetate.

Calibration curves were constructed by plotting peak areas of the compounds/peak areas of internal standard versus concentrations of selected standards. A stock solution was prepared by diluting 99.2, 118.7, 162.3, and 114.8 mg of verbenone, methyl isonicotinate, pheromone, and *p*-anisaldehyde, respectively, in 10 mL ethyl acetate. The standards were obtained by a serial dilution of the stock solution with ethyl acetate and the concentrations are shown in Appendix A. A volume of 0.5 mL of the standards was transferred in GC vials and 0.1 mL of the internal standard was added. The solutions were then injected in the GC–MS following the above-mentioned method. Chromatographic data were analyzed using Masshunter Quantitative Analysis DA 10.2 software (Agilent, Santa Clara, CA, USA). The concentrations of compounds in the eluate from the SPE cartridges were predicted from linear models built with calibration data using the R software (v4.3.1) [30]. The release rates were calculated by multiplying the concentration by the volume of the standards (0.6 mL) and divided by the period of volatile compound collection (50 min). Data from the GC–MS are available in the Appendix A and the R script in Appendix A.

### 2.7. Statistical Analysis

Statistical analysis was conducted with the R software [30]. The behavioral responses of the thrips were analyzed as described in Davidson et al. [21] using a generalized linear model (GLM) set with a quasibinomial distribution and a logit link function. A backward selection of variables was performed to identify the best performing GLM, and each variable’s elimination was assessed by comparing models using a deviance analysis following a Χ^2^ distribution. Given the potential variability in behavioral responses in Y-olfactometer tests with 75 individuals tested per condition, the significance level was set to 0.01 to increase the robustness in distinguishing proportions different from 50%. The release rates of the volatile compounds were analyzed with a linear model (LM) and the best performing LM was determined through an analysis of variance. Raw data are provided in Appendix A and the R script for data analysis and the production of figures can be found in Appendix A.

## 3. Results

### 3.1. Behavioral Responses

According to the behavioral assay, the results demonstrate that the proportion of female thrips attracted to the pheromone were significantly different from 50% for the two higher doses (Figure 2). At the loading of 10 and 100 µg, the proportions of females attracted were 68% and 77.3%, respectively. Conversely, verbenone did not significantly attract females. *p*-Anisaldehyde and methyl isonicotinate significantly attracted females only at a loading of 100 µg, reaching a proportion of 72% and 73.3%, respectively (Figure 2). To evaluate the homogeneity of the three replicates for each treatment, GLMs were compared with and without the replicates as a factor. No significant effect of the replicates was found, as shown by the comparison of GLMs with the analysis of deviance (ANOVA Χ^2^ distribution, *p* = 0.99 for the interaction term and *p* = 0.93 for the intercept). Therefore, data from the three replicates were pooled. The ratio of females found in both arms over the total number tested varied from 74.5% to 89.8%.

### 3.2. Release Rates

We found that the release rates of all four compounds increased linearly with the mass loaded on the filter papers (LM, adj R^2^ = 0.949, *p* < 0.001; Figure 3). The average slopes for each compound were not significantly different (comparison of linear models with ANOVA, *p* = 0.938), the overall slope was 1.049. This means that for an increase in the mass loaded on the filter paper, the release rate of each compound increased by roughly the same magnitude, independently of the compound. This also suggests that the gas-phase saturation was not reached. The intercepts, which represent the starting point for the linear relationship, differed significantly between the four compounds. Methyl isonicotinate and verbenone had significantly higher intercepts than *p*-anisaldehyde and the pheromone (Figure 3). When transformed back from the log scale, the intercepts were 8.7, 8.4, 2.8, and 0.4 ng/min for methyl isonicotinate, verbenone, *p*-anisaldehyde, and the pheromone, respectively. These results suggest that methyl isonicotinate and verbenone evaporated more quickly than *p*-anisaldehyde, while the pheromone had the lowest release rate that was at least 7 times lower than the other compounds. Additionally, we found that the release rates of the pheromone were highly variable compared to the other compounds, especially at 10 and 100 µg loading (Figure 3).

### 3.3. Performance of Volatile Compounds to Attract WFT

We determined the lowest release rate of each compound that produced a behavioral response significantly different from 50%. The pheromone was found to be the most effective compound, with the lowest release rate (4 ng/min) producing a significant attraction of females (Figure 4). On the other hand, *p*-anisaldehyde and methyl isonicotinate required release rates of 412 and 1003 ng/min, respectively, to significantly attract females (Figure 4). The pheromone was found to be 103 and 251 times more effective than *p*-anisaldehyde and methyl isonicotinate, respectively, at attracting female thrips. *p*-Anisaldehyde was roughly two times more effective than methyl isonicotinate. As previously mentioned, verbenone did not produce any significant results.

From the release rates, we can calculate the average gas-phase concentrations of the compounds in the odorant arm during the behavioral assays. The average release rate was divided by the flow rate in the odorant arm (150 mL/min). Table 1 summarizes the mass loaded on the filter paper, the average retransformed log of release rates, the average gas-phase concentrations, and the proportion of attracted females. The pheromone was attractive from a gas-phase concentration of 0.027 ng/mL, whereas the attractivity dropped with gas-phase concentrations below 6.7 and 2.7 ng/mL for methyl isonicotinate and *p*-anisaldehyde, respectively (Table 1). Thus, the gas-phase concentrations of 0.027, 2.7, and 6.7 ng/mL for the pheromone, *p*-anisaldehyde, and methyl isonicotinate, respectively, are the minimal concentrations needed to attract WFT females. 

## 4. Discussion

Our results indicate that the aggregation pheromone neryl (*S*)-2-methylbutanoate is better at attracting female WFT than the kairomones, *p*-anisaldehyde and methyl isonicotinate, in a Y-olfactometer. The minimal gas-phase concentrations of the pheromone required to attract WFT females was at least 100 times lower than that of the other two compounds. Thus, less chemical compound is required to attract WFT when the pheromone is used in devices to deliver volatile compounds under field experiments. For example, the pheromone dispensers use small rubber septa (25 mm diameter, 55 mm long) loaded with 30 µg [31] whereas the dispensers are much larger and loaded with 1 g of methyl isonicotinate [10].

The sensitivity of insects to olfactory stimuli varies with intrinsic and extrinsic factors, and the olfactory system evolved in such a way that sensitivity is required only when the result of a behavioral response increases the fitness [32]. Male moths are very sensitive to sex pheromones emitted by females, as they can detect females from a long distance [26]. To achieve this, they are equipped with a very sensitive olfactory system. For example, males of *Bombyx mori* respond to gas-phase concentrations of the sex pheromone as low as 0.012 fg/mL [33]. In blood-feeding insects, the anthropophilic mosquito *Anopheles gambiae* responds to lactic acid at concentrations of 0.015 ng/mL in a Y-olfactometer. This concentration of lactic acid corresponds to that which emanates from a human hand [34]. The mosquito *Aedes aegypti* responds to CO_2_ levels of 500 ppm (9 µg/mL) above the background, corresponding to the emissions of humans at a distance of several meters [35]. We found that WFT responds to the aggregation pheromone at a gas-phase concentration of 0.027 ng/mL, and to methyl isonicotinate and *p*-anisaldehyde at gas-phase concentrations of 6.7 ng/mL and 2.7 ng/mL, respectively. Measuring the gas-phase concentrations of the pheromones released by WFT males and the concentrations of kairomones released by plants could improve the understanding of how WFT exploits chemical signals from their congeners and plants.

The gas-phase concentrations in the present study were obtained by collecting evaporated compounds over 50 min in controlled laboratory conditions. However, these concentrations may not accurately reflect the temporal and spatial distribution of the pheromone plume emitted by WFT males in the field, as concentrations can vary significantly over shorter times and spatial intervals due to the turbulence of air flow, creating filamentous plumes [26], and due to the fact that the WFT males can intermittently emit the pheromone. Additionally, it is important to note that our Y-olfactometer experiments should be interpreted with caution as WFT in the field are exposed to more complex choices than simply clean air versus one odorant compound. Like other insects, WFT rely on relevant olfactory cues, which are typically mixtures of compounds accompanied by other volatiles that form a background [36]. Measuring the average gas-phase concentrations delivered in behavioral assays can aid in comparing results from different studies conducted in controlled environments. Additionally, it can assist in designing and developing sensors that respond to target concentrations with the appropriate temporal resolution. Furthermore, as models of odor-plume dynamics improve, threshold concentrations will prove valuable in optimizing pest control methods with odor-baited traps.

We found that the log-transformed release rates were linearly correlated to the log-transformed dose loaded on the filter paper. Thus, WFT, in our study, were exposed to gas-phase concentrations linearly correlated to the dose loaded on the filter paper. The ranges of gas-phase concentrations were well below the estimated vapor pressures at 25 °C: 0.5, 4, 18, and 32 Pa for the pheromone, *p*-anisaldehyde, verbenone, and methyl isonicotinate, respectively [37,38]. Note that the vapor pressure of (*S*)-(-)-verbenone was not found; thus, the value for the R-enantiomer was used. Saturation would occur if gas-phase concentrations reached 48, 220, 1091, and 1769 µg/L for each compound, respectively. Vapor pressure can be used to evaluate the volatility of compounds as the evaporation mass flux of a neat compound is a function of the vapor pressure. However, other parameters influence the evaporation mass flux such as the gas-phase composition, temperature, airflow [39], and the release matrix properties. This renders a direct evaluation of the release rates based solely on the vapor pressure difficult. According to a delivery system, filter papers loaded with 1 µL of solution produced release rates repeatable enough to investigate WFT behavioral responses, except for the pheromone. The variation of the pheromone release rates was too high and may contribute to the variation of the behavioral responses of WFT in our study. Rubber septum is an alternative, as it is widely used to deliver pheromones. However, we did not find any data on the release rates of the WFT aggregation pheromone delivered from rubber septa in the literature. The delivery system should be considered in future studies involving WFT behavioral responses to the aggregation pheromone.

While our findings support previous reports on the attraction of WFT to methyl isonicotinate and *p*-anisaldehyde, we observed that WFT only responded to the higher dose of these compounds in our study. This contrasts with the results of two other studies, where WFT was significantly attracted to lower doses of both compounds [17,21]. Although the olfactometer used in our study had similar wind speed, temperature, light intensity, Y-tube tilting angle, filter paper surface, and dose-loading as those used in the previous studies, the Y-tube we used had a larger inner diameter (8 mm instead of 5 mm). Consequently, the air flow was higher (150 mL/min instead of 59 mL/min), possibly resulting in lower gas-phase concentrations in our study. This could partially explain the responses of WFT to the lower doses found in the literature compared to our study. Another difference among studies is the starvation period. In our study, WFT were starved for 1 h whereas, in the previous studies, it was starved for 14 h and 24 h. The starvation period may explain the difference of WFT sensitivity to methyl isonicotinate and *p*-anisaldehyde. Previous research has shown that the responses of WFT to *p*-anisaldehyde, delivered at a single dose (100 µg), increased with the starvation period [40]. Well-fed WFT hardly responded to *p*-anisaldehyde, whereas they clearly responded to it when starved for at least a 4 h period. In insects, the sensitivity of the olfactory system is influenced by many internal factors including hunger [32]. For example, electroantennogram responses of the blood-sucking bug *Rhodnius prolixus* to ammonia were increased by 6–7 times in starved bugs than in fed ones [41]. Behavioral changes due to insect feeding status can be substantial. For instance, *Rhodnius prolixus* become unresponsive to attractants, namely CO_2_, 48 h after blood meals [42]. Similarly, well-fed *Drosophila melanogaster* are much less attracted to food odors from vinegar compared to starved ones [43]. To verify if starvation modulates the sensitivity of WFT to methyl isonicotinate and *p*-anisaldehyde, additional experiments are needed to establish the dose-response curves for each compound with WFT starved for different periods of time. The regulation of olfactory responses in WFT by intrinsic factors, such as feeding status, represents an important issue for the development of effective control methods [44].

In our study, we did not observe any significant responses to (*S*)-(-)-verbenone, which contrasts with the results obtained in a previous study that used a 4-arm olfactometer and a single dose [18]. As discussed earlier, the lack of response to (*S*)-(-)-verbenone by WFT in our study could be due to the starvation period before the experiments. However, field experiments with (S)-(-)-verbenone produced unclear results [45]. In one experiment conducted in England, traps baited with (*S*)-(-)-verbenone significantly caught more thrips than traps baited with the solvent, whereas, in a second experiment conducted in Turkey, the results of the same odor treatment could be hardly distinguished from the solvent control [45]. Recently, a commercial lure, namely Thrips Charm^®^, a mixture of *p*-anisaldehyde and (*S*)-(-)-verbenone significantly increased catches from visual traps in the field [46]. No details about the ratio of both compounds in the commercial lure could be found. 

## 5. Conclusions

The aggregation pheromone is more attractive to WFT in a Y-olfactometer than methyl isonicotinate and *p*-anisladehyde, as WFT responded to gas-phase concentrations of pheromone at least 100 times lower than the ones of the two other compounds. It is important to consider the complexities of the olfactory behavior of thrips, and the limitations of laboratory studies, when interpreting these results. The temporal and spatial distribution of pheromones in the field can be highly variable, and thrips are exposed to more complex olfactory cues in nature than those presented in laboratory experiments. The use of average gas-phase concentrations measured in controlled environments can aid in comparing results from different studies and in designing sensors that respond to target concentrations with the appropriate temporal resolution. Moreover, as models of odor-plume dynamics improve, threshold concentrations will be valuable in optimizing pest control methods with odor-baited traps. Overall, a better understanding of WFT olfactory behavior, and the extrinsic and intrinsic factors that modulate their responses, will be essential for developing effective and sustainable pest management strategies.

## Figures and Tables

**Figure 1 insects-14-00562-f001:**
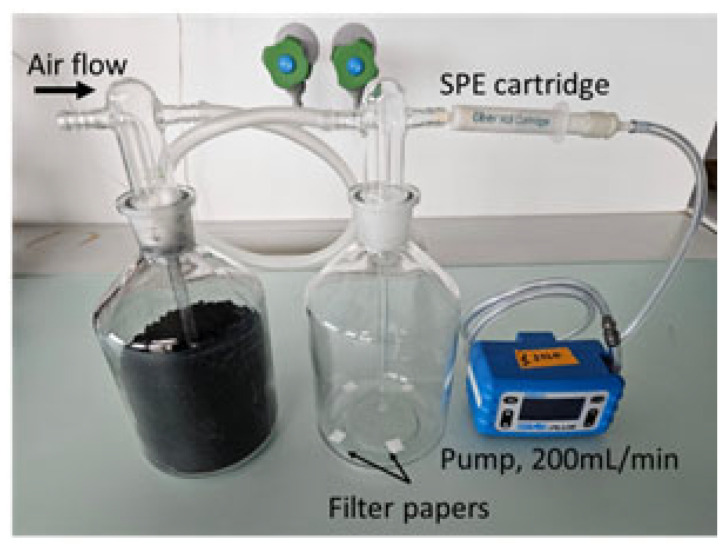
Headspace cell to measure the release rates of volatile compounds loaded on filter papers.

**Figure 2 insects-14-00562-f002:**
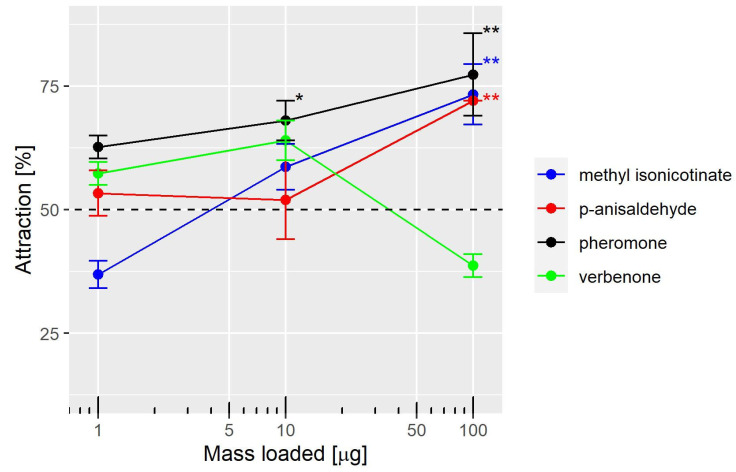
Mean ± standard deviation of the proportions of females that chose the odor arm. The stars show the level of significance for proportions over 75 individuals differing from 50% following a GLM with a binomial distribution: * *p* < 0.01; ** *p* < 0.001.

**Figure 3 insects-14-00562-f003:**
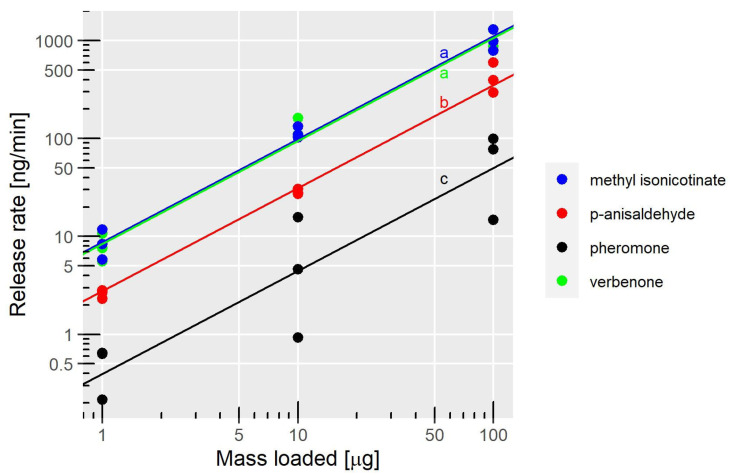
Measurements of the release rates as a function of the dose and the compounds. Solid lines show the linear model for each compound. n = 3 for each treatment. Letters show intercepts that are different at a significant level of *p* < 0.01.

**Figure 4 insects-14-00562-f004:**
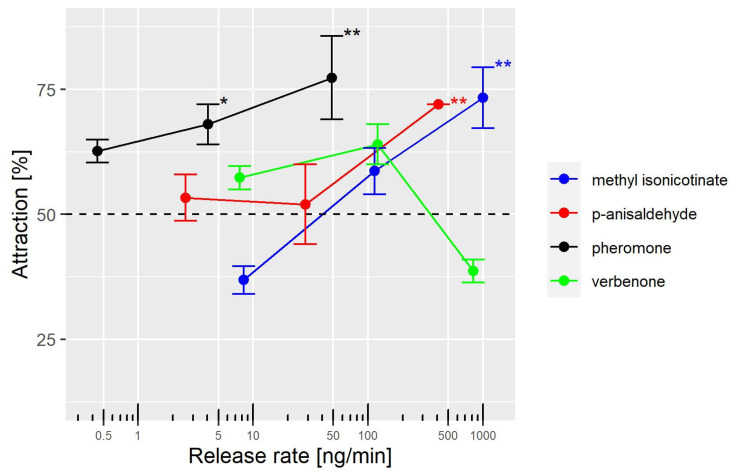
Mean ± standard deviation of the proportions of females attracted as a function of the average release rates. The stars show the level of significance for proportions over 75 individuals differing from 50% following a GLM with a binomial distribution: * *p* < 0.01; ** *p* < 0.001.

**Table 1 insects-14-00562-t001:** Mass loaded on the filter paper with the corresponding average retransformed release rates measured in the headspace cells, the corresponding calculated gas-phase concentration and the percentage of female thrips that chose the odorant arm.

Compounds	Mass Loaded [µg]	Release Rate [ng/min]	Gas-Phase Concentration [ng/mL]	Attraction [%]
Pheromone	1	0.4	0.003	62.7
	10	4	0.027	68
	100	48	0.322	77.3
Verbenone	1	8	0.051	57.3
	10	122	0.812	64
	100	826	5.5	38.7
*p*-Anisaldehyde	1	3	0.017	53.3
	10	28	0.19	52
	100	412	2.7	72
Methyl isonicotinate	1	8	0.055	36.9
	10	114	0.761	58.7
	100	1003	6.7	73.3

## Data Availability

Data supporting the reported results can be found in the Appendix A section.

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
