# Peer review of "Attraction of *Frankliniella occidentalis* Females towards the Aggregation Pheromone Neryl (*S*)-2-Methylbutanoate and Kairomones in a Y-Olfactometer"

_insects, 2023, doi:10.3390/insects14060562_

Round 1
Reviewer 1 Report
Dear Alyssa Kang
Assistant Editor
Insects Editorial Office
Attracting performance of the pheromone neryl (S)-2-methyl-2 butanoate and kairomones towards the female western flower thrips (Frankliniella occidentalis) in a Y-olfactometer
Charles JF Chappuis,*, Marilyn Cléroux, Corentin Descombes, Yannick Barth, François Lefort
The manuscript is well written and structured, very easy to read and assimilate. However, from my point of view, it is an article that contributes little to the generation of new knowledge, I consider that the evaluation of the aggregation pheromone and two kairomonal compounds in a type Y olfactometer is too little information for a scientific article in a high-quality journal such as Insects; perhaps if the authors added relevant information such as the electroanthenographic response or behavior in the wind tunnel or greenhouse to the evaluated attractants, the manuscript could be considered more complete and not as it currently is. Therefore, I consider that this work is not yet ready to be published in the journal.
Additionally, I have made a series of comments, suggestions and recommendations, which I list below. I hope that my suggestions and observations contribute to improving the quality of this manuscript.
Specific comments and suggestions
L28 = …SPE… Put its meaning the first time it is mentioned.
L172 = … in microcosms (60cm X 60cm X 60cm)… Please add what material was the microcosm made of for breeding thrips?
L176 = … genetically identified… Please attach the access number, percentage identity, phylogenetic tree comparing the sequence obtained from the species with those previously reported in the gene bank.
L190 = …10cm/s with the valve. The total airflow was 300ml/min, 150ml/min in each… The authors first use cm/s to measure flow and then use ml/min. This criterion needs to be clarified or unified.
L199 = … Microbox…and microcosms (L 172) It´s are the same? describe what they were made of.
L215 = … (Figure 1)… I would like the authors to put the entire image of the experiment and not limit themselves to the bomb only, so you could see the cartridges where the volatiles were collected.
L216-220 = … The pheromone, methyl isonicotinate and verbenone were diluted in hexane at concentrations of 1mg/mL, 10mg/mL and 100mg/mL. P-anisaldehyde was diluted in water at the same con- centrations mentioned above… This information was already mentioned in lines 191-196.
247-248 = …Nine 500μL standards were prepared …from 9.92ng/mL to 162.3μg/mL (Table S1). It is very confusing and extremely difficult to follow the text with the data in table S1. The data does not match the data in the table. In addition, the measurement units are different in the table and in the text, which makes it even more difficult to relate the text to the table and therefore makes it difficult to read and, above all, to understand both. For example, concentrations from 9.92ng/mL to 162.3μg/mL are not seen in the table (Table S1). I suggest modifying and unifying the information for a clear understanding of the data.
L315-317 = When the authors say …”We compared the lowest average release rate of each compound that produced a behavioral response significantly different from 50%.” What exactly are the authors referring to? since in figure 4 the X-axis reads 0, 1, 2, 3 ng/min? This point needs to be clarified.
L317-318 = … The pheromone was found to be the most effective compound, with the lowest release rate (4 ng/min) producing a significant attraction of females (Figure 4)…. In this text the authors mention that the pheromone was the most attractive for the female thrips with the lowest release rate (4 ng/min). Two very important points must be mentioned here. First, in figure 4 the data can only be seen up to 3 ng/min on the X axis. Second, in figure 4 it can be seen that the best captures with the pheromone do not exceed 2 ng/min. These points need to be clarified and corrected.
L318-320 = … On the other hand, p-anisaldehyde and methyl isonicotinate required release rates of 412 and 1003 ng/min, respectively, to significantly attract females (Figure 4)…. I insist again, the data in the text (412 and 1003 ng/min) do not coincide with those at the bottom of figure 4 (Log of release rate ng/min; 0, 1, 2, 3). Perhaps it is due to the scale used in the X axis. If this is the case, the authors must clarify or modify this information so that the text matches the figure. This point needs to be clarified and corrected.
L365 = …Aedes aegypti… italicize
Reviewer 2 Report
Article, which I have received for review presents the results of the laboratory studies on the behaviour of Frankliniella occidentalis as a response to the application of four semiochemicals.
Among them kairomones: methyl isonicotinate and p-anisaldehyde are the most often studied substances, and the data about using them against F. occidentalis may be found in many scientific articles. The third kairomone (S)-(−)-verbenone and the pheromone - neryl (S)-2-methyl butanoate also have been studied previously by other scientists.
Introduction, Results and Discussion are not objectionable to me.
Most elements for change are in References - the names of cited journals should be written accordance with the guidelines of Insects journal.
All my comments I put in the text.
Since I am not a chemist, I cannot competently evaluate parts: Chemicals, Pheromone synthesis and GC-MS/MS analysis of the Materials and Methods chapter.

Author Response
General comment from the authors: We are grateful to the reviewer for its useful comments and critics which allowed to improve our manuscript. Please find below the specific answers to the reviewers’ comments.
Response to REVIEWER 2 comments
Article, which I have received for review presents the results of the laboratory studies on the behaviour of Frankliniella occidentalis as a response to the application of four semiochemicals.
Among them kairomones: methyl isonicotinate and p-anisaldehyde are the most often studied substances, and the data about using them against F. occidentalis may be found in many scientific articles. The third kairomone (S)-(−)-verbenone and the pheromone - neryl (S)-2-methyl butanoate also have been studied previously by other scientists.
Introduction, Results and Discussion are not objectionable to me.
Point 1: Most elements for change are in References - the names of cited journals should be written accordance with the guidelines of Insects journal.
Response 1: References were checked and the corrections were made according to the reviewer’s comments.
Point 2: All my comments I put in the text.
Response 2: We made all the corrections mentioned by the reviewer.
Since I am not a chemist, I cannot competently evaluate parts: Chemicals, Pheromone synthesis and GC-MS/MS analysis of the Materials and Methods chapter.
Reviewer 3 Report
The manuscript entitled `Attracting performance----------in a Y-olfactometer’ contains valuable information for developing sustainable management program against western flower thrips. This thrips is a serious pest of a large number of crops mostly for it virus transmission capability. The manuscript is well written and supported by many references. Th introduction and discussion is relevant and well supported. I have some minor comments for the authors. After the minor revision, the manuscript can be accepted for publication in the `Insects.’ The comments are shown below:
Simple summary
Line 13: Replace `insects smell’ with `insects olfactory perception’.
Line 14: use `manage’ instead of `control’.
Abstract
Line 23: Replace `A’ with `An’. On the same line, insert `of’ after `understanding’.
Line 25: use` to estimate’ instead of `estimated’.
Line 29: Replace `We show’ with `We observed’.
Introduction
Line 41: Replace `a formidable’ with `an economic.’
Line 87: Use `in’ instead of `on’.
Line 88: Use `in’ for `on’.
Lines 98-99: Reword sentence, `Insects respond-------matrix.’ The meaning is not clear.
Materials and Methods
Line 139: Suggested to use `n-hexane (99%) used in this study were’ instead of `n-hexane (99%), and’, also provide complete address of `Carl Roth’.
Line 144: Provide complete address of Sigma-Aldrich
Line 145: Spell out abbreviation at the first citation, e.g.; SPE, OASIS, HL and Waters.
Line 151: Source of round -bottom flask,
Line 152: Gently stirred for how long?
Line 153: How long the round-bottom flask was in the ice bath?-mention.
Line 154: Replace `during’ with `for’.
Line 154: What kind of paper-mention.
Lines 290-293: We used-------filter paper. -Move these lines to M & M.
Lines 314-316: Move to M & M
The manuscript entitled `Attracting performance----------in a Y-olfactometer’ contains valuable information for developing sustainable management program against western flower thrips. This thrips is a serious pest of a large number of crops mostly for it virus transmission capability. The manuscript is well written and supported by many references. Th introduction and discussion is relevant and well supported. I have some minor comments for the authors. After the minor revision, the manuscript can be accepted for publication in the `Insects.’ The comments are shown below:
Simple summary
Line 13: Replace `insects smell’ with `insects olfactory perception’.
Line 14: use `manage’ instead of `control’.
Abstract
Line 23: Replace `A’ with `An’. On the same line, insert `of’ after `understanding’.
Line 25: use` to estimate’ instead of `estimated’.
Line 29: Replace `We show’ with `We observed’.
Introduction
Line 41: Replace `a formidable’ with `an economic.’
Line 87: Use `in’ instead of `on’.
Line 88: Use `in’ for `on’.
Lines 98-99: Reword sentence, `Insects respond-------matrix.’ The meaning is not clear.
Materials and Methods
Line 139: Suggested to use `n-hexane (99%) used in this study were’ instead of `n-hexane (99%), and’, also provide complete address of `Carl Roth’.
Line 144: Provide complete address of Sigma-Aldrich
Line 145: Spell out abbreviation at the first citation, e.g.; SPE, OASIS, HL and Waters.
Line 151: Source of round -bottom flask,
Line 152: Gently stirred for how long?
Line 153: How long the round-bottom flask was in the ice bath?-mention.
Line 154: Replace `during’ with `for’.
Line 154: What kind of paper-mention.
Lines 290-293: We used-------filter paper. -Move these lines to M & M.
Lines 314-316: Move to M & M
Author Response
General comment from the authors: We are grateful to the reviewer for its useful comments and critics which allowed to improve our manuscript. Please find below the specific answers to the reviewers’ comments.
Response to REVIEWER 3 comments
The manuscript entitled `Attracting performance----------in a Y-olfactometer’ contains valuable information for developing sustainable management program against western flower thrips. This thrips is a serious pest of a large number of crops mostly for it virus transmission capability. The manuscript is well written and supported by many references. Th introduction and discussion is relevant and well supported. I have some minor comments for the authors. After the minor revision, the manuscript can be accepted for publication in the `Insects.’ The comments are shown below:
Simple summary
Point 1: Line 13: Replace `insects smell’ with `insects olfactory perception’.
Response 1: Change has been made Line 13.
Point 2: Line 14: use `manage’ instead of `control’.
Response 2: Change has been made Line 14.
Abstract
Point 3: Line 23: Replace `A’ with `An’. On the same line, insert `of’ after `understanding’.
Response 3: Changes have been made Line 23.
Point 4: Line 25: use` to estimate’ instead of `estimated’.
Response 4: Change has been made Line 25.
Point 5: Line 29: Replace `We show’ with `We observed’.
Response 5: Change has been made Line 29.
Introduction
Point 6: Line 41: Replace `a formidable’ with `an economic.’
Response 6: Change has been made Line 42.
Point 7: Line 87: Use `in’ instead of `on’.
Response 7: Change has been made Line 88.
Point 8: Line 88: Use `in’ for `on’.
Response 8: Change has been made Line 89.
Point 9: Lines 98-99: Reword sentence, `Insects respond-------matrix.’ The meaning is not clear.
Response 9: We reworded the sentence Line 100-102 to make it clearer: “Insects respond to a part of the amount of compounds that evaporate and are carried by the air, not to the total amount of compounds that is loaded on a matrix such as a filter paper”.
Materials and Methods
Point 9: Line 139: Suggested to use `n-hexane (99%) used in this study were’ instead of `n-hexane (99%), and’, also provide complete address of `Carl Roth’.
Response 9: Change was made Line 142-143 and City, state and country was added.
Point 10: Line 144: Provide complete address of Sigma-Aldrich
Response 10: City, state and country was added Line 148.
Point 11: Line 145: Spell out abbreviation at the first citation, e.g.; SPE, OASIS, HL and Waters.
Response 11: SPE was spelled in the abstract out (Solid-phase extraction). OASIS HLB are the products name of the Waters company.
Point 12: Line 151: Source of round -bottom flask,
Response 12: This is common laboratory material. This material can be purchased from a lot of different vendors. . Here is an example of a company that sell this kind of flask: https://fr.vwr.com/store/product/2103924/vwr-ballons-fond-rond-avec-col-rode-normalise
Point 13: Line 152: Gently stirred for how long?
Response 13: we reworded Line 156 to make it more clear. During addition the mixture was stirred. This step lasted 30min.
Point 14: Line 153: How long the round-bottom flask was in the ice bath?-mention.
Response 14: 30min. DCC was added during 30min. The flask was in the ice bad during this procedure and the content stirred during the same period. We think the previous reworded sentence should make it more clear.
Point 15: Line 154: Replace `during’ with `for’.
Response 15: Change was made Line 158.
Point 16: Line 154: What kind of paper-mention.
Response 16: The following was added in the text Line 159: Whatman™ 589/2 (150mm in diameter, Sigma, Buchs, Saint-Gallen, Switzerland)
Point 17: Lines 290-293: We used-------filter paper. -Move these lines to M & M.
Response 17 : We removed this part as the main point (similar conditions) is mentioned in the introduction.
Point 18: Lines 314-316: Move to M & M
Response 18: We removed this part as it brought nothing to the text.
Round 2
Reviewer 1 Report
Dear Alyssa Kang
Assistant Editor
Insects Editorial Office
Attracting performance of the pheromone neryl (S)-2-methyl-2 butanoate and kairomones towards the female western flower thrips (Frankliniella occidentalis) in a Y-olfactometer
insects-2382961
Charles JF Chappuis,*, Marilyn Cléroux, Corentin Descombes, Yannick Barth, François Lefort
Regarding the revised version of the manuscript "Attracting performance of the pheromone neryl (S)-2-methyl-2 butanoate and kairomones towards the female western flower thrips (Frankliniella occidentalis) in a Y-olfactometer", forwarded by the authors Charles JF Chappuis, Marilyn Cléroux, Corentin Descombes, Yannick Barth, François Lefort, my point of view is the following:
Although the authors made almost all the observations and suggestions that I proposed, they lacked the most significant and important for the story of this article to be a complete story, which is to carry out behavioral bioassays in a flight tunnel or in a greenhouse or bioassays of electroantennagraphy; therefore I consider that the manuscript is still not ready to be published yet.
I await my suggestions to help improve the quality of this manuscript.
Kind regards
